A fossil unicorn crestfish (Teleostei, Lampridiformes, Lophotidae) from the Eocene of Iran

Davesne Donald donald.davesne@earth.ox.ac.uk
Department of Earth Sciences, University of Oxford , Oxford , United Kingdom
Anquetin Jérémy
Electronic publication date: 2017 Jun 28
Publication date: 2017
Volume: 5
Electronic Location ID: e3381
Received 2017 Apr 12; Accepted 2017 May 5
Copyright: ©2017 Davesne
Copyright year: 2017
Copyright holder: Davesne
License: This is an open access article distributed under the terms of the Creative Commons Attribution License, which permits unrestricted use, distribution, reproduction and adaptation in any medium and for any purpose provided that it is properly attributed. For attribution, the original author(s), title, publication source (PeerJ) and either DOI or URL of the article must be cited.
License URL: https://creativecommons.org/licenses/by/4.0/

Keywords: Eocene, Iran, Teleostei, Acanthomorpha, Lampridiformes, Taeniosomi, Lophotidae, Taxonomy, Fossil record

Funding: Natural Environment Research Council NE/J022632/1 Leverhulme Trust RPG- 2016-168 The author was supported by the Natural Environment Research Council (grant NE/J022632/1) and by the Leverhulme Trust (grant RPG- 2016-168). The funders had no role in study design, data collection and analysis, decision to publish, or preparation of the manuscript.

==============================
Lophotidae, or crestfishes, is a family of rare deep-sea teleosts characterised by an enlarged horn-like crest on the forehead. They are poorly represented in the fossil record, by only three described taxa. One specimen attributed to Lophotidae has been described from the pelagic fauna of the middle-late Eocene Zagros Basin, Iran. Originally considered as a specimen of the fossil lophotid †Protolophotus, it is proposed hereby as a new genus and species †Babelichthys olneyi, gen. et sp. nov., differs from the other fossil lophotids by its relatively long and strongly projecting crest, suggesting a close relationship with the modern unicorn crestfish, Eumecichthys. This new taxon increases the diversity of the deep-sea teleost fauna to which it belongs, improving our understanding of the taxonomic composition of the early Cenozoic mesopelagic ecosystems.

Introduction

Lampridiforms are strange spiny-rayed teleosts, found in mesopelagic environments in every ocean of the world (Olney, Johnson & Baldwin, 1993; Olney, 2002). Their most famous representatives are the endothermic opah (Lampris guttatus) and the gigantic, serpentine oarfish (Regalecus glesne), the longest known teleost. Along with these iconic taxa, lampridiforms include equally weird ribbon-like and elongate animals, characterized by a silver-coloured skin and long, bright red fins: the taeniosomes. The 15–18 extant species of the clade Taeniosomi include oarfishes (Regalecidae), ribbonfishes (Trachipteridae), the tapertail (Radiicephalidae) and Lophotidae, the crestfishes (Regan, 1907; Walters & Fitch, 1960; Olney, 1984; Roberts, 2012). Lophotids are characterized by unique anatomical structures, such as an ink gland (Walters & Fitch, 1960; Honma, Ushiki & Takeda, 1999) not found anywhere else in teleosts (except in the closely related radiicephalids; Harrisson & Palmer, 1968). The most conspicuous osteological feature of lophotids is a well-developed horn-like crest, formed by an anteriorly projecting expansion of the frontal and supraoccipital bones of the cranium (Oelschläger, 1979; Oelschläger, 1983; Olney, Johnson & Baldwin, 1993). This crest is closely associated with the anterior pterygiophores supporting the dorsal fin, and as a result, the dorsal fin expands over, and sometimes anterior to the cranium. Lophotids are represented in modern fauna by one to three Lophotus species and by the unicorn crestfish, Eumecichthys fiski (Walters & Fitch, 1960; Craig, Hastings & Pondella, 2004). Their fossil record consists in at least three monotypic genera (Bannikov, 1999; Carnevale, 2004). The present article is a revision of an anatomically distinctive fossil specimen attributed to Lophotidae. Arambourg (1943) and Arambourg (1967) first described the specimen from a rich late Eocene fauna located near Ilam, Zagros Basin, Iran. The Ilam fauna comprises numerous representatives of teleost taxa such as Beryciformes, Gadiformes, Ophidiiformes and Stomiiformes, typical of the modern deep-sea pelagic environments (Arambourg, 1967; Afsari et al., 2014; Přikryl, Brzobohatý & Gregorová, 2016).

The taxonomic status of the lophotid specimen studied here is currently unclear (Walters, 1957; Oelschläger, 1979; Bannikov, 1999), and it lacks a proper anatomical description. Given the rarity of fossil material attributed to taeniosome lampridiforms, a detailed description and revised taxonomy of this material is needed in order to improve our understanding of the morphological evolution and fossil record of this peculiar group.

Material and Methods

Taxonomic status of the material

The material described herein, MNHN.F.EIP11 (Figs. 1 and 2), was discovered during excavations near Ilam (Zagros Basin, Western Iran) by Camille Arambourg in 1938–1939. The specimen was chosen to be the paratype of †Lophotes elami (Arambourg, 1943), along with the holotype MNHN.F.EIP10 (Fig. 3). On the basis of osteological differences from extant lophotids, such as the well-ossified pelvic girdle in a ventral position observed in the holotype of †Lophotes elami, Walters (1957) assigned it to a distinct new genus †Protolophotus (Fig. 3). Oelschläger (1979) later proposed that MNHN.F.EIP11 differs sufficiently from MNHN.F.EIP10 to be classified in a different genus. He related the specimen to the extant Eumecichthys and gave it the name †‘Protomecichthys’. However, the genus †‘Protomecichthys’ lacks both a designated type species and a formal description. Thus, it fails to meet the requirements of Article 13.3 of the International Code of Zoological Nomenclature (International Commission on Zoological Nomenclature, 1999) and should be considered a nomen nudum (Bannikov, 1999).

Figure 1 †Babelichthys olneyi, gen. et sp. nov., holotype.

(A) MNHN.F.EIP11d. (B) counterpart MNHN.F.EIP11g. Scale bars = 20 mm.

Figure 2 †Babelichthys olneyi, gen. et sp. nov. holotype MNHN.F.EIP11d.

Photograph (detail of the head) and interpretative drawing. Legend: achy, anterior ceratohyal; bra, branchiostegal; bsp, basisphenoid; den, dentary; dfr, dorsal-fin ray; dhhy, dorsal hypohyal; dpt, dorsal-fin pterygiophore; enpt, endopterygoid; fr, frontal; hyo, hyomandibula; iop, interopercle; lac, lachrymal; let, lateral ethmoid; mpt, metapterygoid; mx, maxilla; osp, orbitosphenoid; pal, palatine; pchy, posterior ceratohyal; pmx, premaxilla; pop, preopercle; psp, parasphenoid; qu, quadrate; soc, supraoccipital; soc-sp, spine of the supraoccipital; spl, splint of the first dorsal-fin ray; vhhy, ventral hypohyal; vo, vomer. Scale bar = 10 mm.

Comparative material

†Eolophotes lenis, PIN 1413/86; Eumecichthys fiski, USNM 164170 (radiographs); Lophotus lacepede, NHMUK 1863.8.27.1 (radiographs); †Oligolophotes fragosus, PIN 3363/121; †Protolophotus elami, MNHN.F.EIP10.

Nomenclatural acts

The electronic version of this article in Portable Document Format (PDF) will represent a published work according to the International Commission on Zoological Nomenclature (ICZN), and hence the new names contained in the electronic version are effectively published under that Code from the electronic edition alone. This published work and the nomenclatural acts it contains have been registered in ZooBank, the online registration system for the ICZN. The ZooBank LSIDs (Life Science Identifiers) can be resolved and the associated information viewed through any standard web browser by appending the LSID to the prefix http://zoobank.org/. The LSID for this publication is: urn:lsid:zoobank.org:pub:B677BA4F-CCF4-4678-A8A8-502F059704D2. The online version of this work is archived and available from the following digital repositories: PeerJ, PubMed Central and CLOCKSS.

Methods

The specimen was examined with a stereomicroscope equipped with a camera lucida drawing arm. The interpretative drawing was produced with Adobe Illustrator CS6 from the camera lucida drawings and from photographs. Measurements were taken with a compass or with the software ImageJ 1.5 from radiographs; angles were also measured with ImageJ. The method for estimating the degree of projection of the crest is modified from Craig, Hastings & Pondella (2004): it is based on the angle between the straight line from the tip of the crest to the proximal end of its anterior margin (instead of the tip of the upper jaw, due to varying jaw positions in fossils) and the vertical line drawn perpendicular to the main axis of the parasphenoid (instead of the vertebral column, not preserved in MNHN.F.EIP11). The relative length of the crest is the ratio between the crest length (distance between the tip of the crest and the proximal end of its anterior margin) and the head length without the crest (from the anterior margin of the ethmoid region to the posterior margin of the neurocranium). All extinct taxa are indicated with a dagger (†).

Results

Systematic palaeontology

TELEOSTEI Müller, 1845	
ACANTHOMORPHA Rosen, 1973	
Order LAMPRIDIFORMES Goodrich, 1909	
Suborder TAENIOSOMI Gill, 1885	
Family LOPHOTIDAE Bonaparte, 1845	
Genus †Babelichthys gen. nov.	
urn:lsid:zoobank.org:act:86986E5E-5FFF-465D-A987-E475FBF02966	
(Figs. 1 and 2)	

Etymology. Hellenization of the name of the “Babel fish”, the teleost-like, ear-dwelling, polyglot extra-terrestrial species from Douglas Adams’ 1979 book The Hitchhiker’s Guide to the Galaxy, in reference to the very peculiar, almost alien-like, appearance of the genus.

Figure 3 †Protolophotus elami, holotype MNHN.F.EIP10d.

Scale bar = 20 mm.

Type and only species. †Babelichthys olneyi, sp. nov.

Diagnosis. A lophotid differing from †Eolophotes, Lophotus, †Oligolophotes and †Protolophotus by the relatively longer, strongly projecting crest; and from Eumecichthys by the relatively shorter, deeper and less strongly projecting crest, with strongly expanded anterior dorsal-fin pterygiophores.

†Babelichthys olneyi sp. nov.

urn:lsid:zoobank.org:act:D2540D1F-F169-40DE-B910-7302810615E7	
(Figs. 1 and 2)	
	
1943 †Lophotes elami Arambourg, p. 287, pl. X, fig. 1	
1957 †Protolophotus elami Walters, p. 60	
1967 †Protolophotes elami Arambourg, pl. VI, fig. 1	
1979 †Protomecichthys sp. Oelschläger, p. 354, fig. 11 (nomen nudum)	

Holotype. MNHN.F.EIP11d/g, almost complete articulated cranium and anterior portion of the dorsal fin, in part and counterpart (Figs. 1 and 2). This is the only specimen known for the genus and species.

Etymology. Species named in honour of the late John E. Olney, in recognition of his work on the anatomy and ontogeny of Lampridiformes.

Type locality and horizon. Near Ilam, Zagros Basin, Western Iran. This teleost fauna, part of the Pabdeh Formation, was erroneously aged Cretaceous by Priem (1908), and Rupelian (Oligocene) by Arambourg (1943) and Arambourg (1967). It is more accurately middle to late Eocene in age (Afsari et al., 2014; and references therein).

Diagnosis. As for the genus.

Anatomical description

MNHN.F.EIP11 consists only of the head of the animal, along with the associated anterior portion of the dorsal fin. The specimen is mostly articulated, except for the left ventral portion of the hyoid arch that is upturned and preserved ventral to the rest of the cranium. The limits of most bones are poorly preserved, probably due to their low degree of mineralization in life as is the case in modern taeniosome lampridiforms.

Measurements

Total head length: 104 mm; head length (without the crest): 44 mm; crest length (anterior margin): 51.5 mm; head depth: 25.5 mm; orbit diameter: 23 mm.

Neurocranium

The neurocranium of MNHN.F.EIP11 is highly modified. The frontal develops a dorsal lamina that projects anterior to the jaws. Throughout approximately its anterior half, it is in contact with an enlarged laminar process of the supraoccipital, delimited dorsally by a strong supraoccipital spine. Together, they form a conspicuous “crest”, long and strongly projecting anteriorly (at an angle of 64.5°). Alone, the crest contributes to 58% of total head length.

The frontal makes up approximately 60% of the anterior margin of the crest. Both the frontal and the supraoccipital show radial ornamentation on the crest; it radiates from the posterior end of the frontal and the distal tip of the supraoccipital. The supraoccipital spine borders the dorsal margin of the bone, and narrows towards the tip.

The ethmoid region is poorly preserved, with an probable enlarged lateral ethmoid that hides the mesethmoid. An enlarged lachrymal is nested in the antero-ventral corner of the orbit; it is parallel to the parasphenoid ventrally, and curves dorsally along the posterior edge of the lateral ethmoid. The orbitosphenoid runs along the dorsal margin of the orbit and has a conspicuous process pointing ventrally. The posterior wall of the orbit is delimited ventrally by a robust and straight basisphenoid. Otherwise, the sphenoid, otic and occipital regions are too poorly preserved to distinguish the individual bones. The parasphenoid is robust and slightly curves dorsally at its anterior end. The junction between the parasphenoid and the vomer is not discernable. There is no evidence of vomerine teeth.

Jaws

The premaxilla is relatively small, with a well-developed ascending process, and a barely visible alveolar process. The maxilla bears a conspicuous and pointed process at its antero-dorsal end. The posterior end is expanded dorsoventrally, forming a rounded lamina. Neither the premaxilla nor the maxilla bear visible teeth. There is no evidence of a supramaxilla. The anterior margin of the dentary, slightly concave and bearing no visible teeth, forms a strong angle with the ventral margin. The posterior margin of the dentary forms an interosseous space with the anguloarticular, which is mostly hidden by overlaying bones.

Suspensorium and hyoid arch

Only the proximal, single-headed articulation of the hyomandibula is clearly visible; the distal end of the bone seems to be preserved in close association with the metapterygoid. The latter is roughly triangular and is one of the best preserved bones of the suspensorium. The symplectic is rod-like, narrows slightly anteriorly and inserts in a notch on the postero-ventral margin of the quadrate. The triangular quadrate bears an antero-ventral condyle that articulates with the anguloarticular. The anterior portion of the suspensorium is poorly preserved, and it is difficult to outline the limits of the endopterygoid, ectopterygoid and palatine bones. The dorsal and posterior portions of the endopterygoid are preserved, suggesting that the bone forms two laminae, the dorsal one along the parasphenoid, and the ventral one contacting both the quadrate and the metapterygoid.

Both the left and right ventral hyoid arches are visible. One is preserved in life position: its posterior end overlaps the operculum, but its dorsal margin is hidden by the lower jaw, suggesting it corresponds to the right ventral hyoid arch. The left one is displaced and upturned, and lies ventral to its counterpart. The posterior ceratohyal is triangular and articulates with the anterior ceratohyal with an interdigitated suture. The anterior ceratohyal shows a strong ventral concavity at midlength; its dorsal margin is much less concave. The anterior end of the anterior ceratohyal forms a rounded condyle, over which the curved ventral hypohyal articulates. The dorsal hypohyal lies dorsally over the anterior ceratohyal. There are six branchiostegals: the anterior two are shorter and articulate with the anterior ceratohyal at the level of its ventral concavity; the four others articulate more posteriorly (due to the faint distinction between both ceratohyals, it is difficult to determine on which one they articulate); they are very long (the posteriormost being the longest) and curved posteriorly over the ventral margin of the interopercle. The branchiostegals of the left hyoid arch are disarticulated.

Opercular series

The preopercle is wide and angled at mid-length. The interopercle is an elongate bone rounded at its extremity that forms the ventral margin of the opercular series. It has a smooth ventral margin, closely associated with the posterior branchiostegals. The potential presence of parts of the opercle, in contact with the preopercle, is unclear.

Dorsal fin and supports

The dorsal fin is only partially preserved, with only the most anterior pterygiophores and dorsal-fin rays visible. Its most striking feature is the extremely elongated and enlarged first dorsal-fin ray, which is 10 times as wide as the more posterior fin rays (at their base and greatest width). It does not bifurcate distally, lacks any visible segmentation, and a groove runs throughout its length. A rounded splint protrudes at its anterior base; it is unclear whether it constitutes a separate dorsal-fin element or not. Fifteen other dorsal-fin rays are preserved posteriorly. Their distal end is missing in most cases, but they all seem to be of a similar length, except for the second and third dorsal-fin rays that are noticeably longer. They do not bifurcate distally, and no segmentation is clearly visible.

Ten dorsal-fin pterygiophores are unambiguously preserved. They are strongly inclined anteriorly, which causes the dorsal fin to originate at the tip of the crest, and to run along the entire head of the animal. The first two dorsal-fin pterygiophores are greatly enlarged and in close contact with the crest. Both also show a conspicuous flange at their posterior margin. The first pterygiophore is narrow posteriorly, where it does not contact the supraoccipital, and widens in its distal end. The second one is much wider and slightly narrows at its distal extremity. It is in close contact with the first pterygiophore throughout its entire length. The third and fourth preserved pterygiophores are in close contact with the second one throughout almost all of their lengths. The more posterior pterygiophores have a mostly straight shaft that curves slightly at its distal extremity. The most posterior ones are less inclined than the anterior ones. The proximal ends of all preserved pterygiophores converge at the same point: the base of the crest—thus they insert anterior to the (not preserved) first neural spine. The elongated and enlarged first dorsal-fin ray inserts on the first pterygiophore. It is unclear if the rays two to eight insert on pterygiophores that are mostly hidden or not preserved, or in supernumerary association with the enlarged second pterygiophore. The rays 9–16 each insert serially on a corresponding pterygiophore.

Discussion

Taxonomic justification

Oelschläger (1979) proposed that MNHN.F. EIP11 is different enough anatomically from the other lophotids, fossil and extant, to justify its attribution to a new genus. Indeed, it differs from the holotype of †Protolophotus, found in the same geological levels, by the relative development of the crest. In MNHN.F.EIP11, the crest is projecting anteriorly with an angle of 64.5°, and the ratio between the lengths of the crest’s anterior margin and of the head without the crest is of 1.17 to 1. In the holotype of †Protolophotus, MNHN.F.EIP10 (Fig. 3), the anterior margin of the crest is almost vertical (degree of projection: 20°), and it is relatively shorter (margin of the crest/head length without the crest: 0.67/1). MNHN.F.EIP11 also shows a much stronger first dorsal-fin ray, and its two anterior dorsal-fin pterygiophores are much more developed. Body size is known to affect relative crest size and degree of projection in extant Lophotus (Craig, Hastings & Pondella, 2004), which could be misleading when trying to differentiate taxa based on crest morphology. However, this bias can probably be ruled out in the case of MNHN.F.EIP11 and MNHN.F.EIP10: both individuals have similar head lengths without the crest (42 and 44 mm, respectively), suggesting that they are at a similar growth stage. It then seems that classifying MNHN.F.EIP11in a different genus and species, †Babelichthys olneyi, is justified from a morphological point of view.

Systematic position

Babelichthys as a taeniosome lampridiform

The monophyly of Lampridiformes (excluding Stylephorus, sensu Nelson, Grande & Wilson, 2016) is well-supported by molecular phylogenetic analyses (Wiley, Johnson & Dimmick, 1998; Miya et al., 2007; Betancur et al., 2013; Near et al., 2013) and by numerous morphological features (Olney, Johnson & Baldwin, 1993; Davesne et al., 2014; Davesne et al., 2016; Delbarre, Davesne & Friedman, 2016). Several of these character states are unambiguously found in †Babelichthys: the premaxilla and dentary are toothless, the frontal and the supraoccipital are both involved in a sagittal crest, the anterior ceratohyal forms a condyle that articulates with the ventral hypohyal, and the first dorsal-fin pterygiophore inserts anterior to the neural spine of the first abdominal vertebra.

The phylogenetic studies that explore lampridiform intrarelationships with a sufficient sampling all recover a monophyletic Taeniosomi (Wiley, Johnson & Dimmick, 1998; Grande, Borden & Smith, 2013; Martin, 2015). The taeniosome character states found in †Babelichthys include the absence of supraneurals, and anterior dorsal-fin pterygiophores that are enlarged and inclined over the neurocranium (Olney, Johnson & Baldwin, 1993). †Babelichthys then clearly shows a character state combination that confirms its identification as a taeniosome lampridiform.

Position within Lophotidae

Olney, Johnson & Baldwin (1993) proposed that the enlarged supraoccipital process, projecting anteriorly over the frontals (forming the “crest” as described herein) and supporting the first dorsal-fin pterygiophore, is a synapomorphy of Lophotidae. Since it is not found elsewhere in lampridiforms, this character confirms the attribution of †Babelichthys to Lophotidae. It has to be noted that in the yet unpublished phylogenetic analysis of Martin (2015), the monophyly of Lophotidae is ambiguous, with one parsimonious tree finding Lophotus more closely related to the other taeniosomes than to Eumecichthys, while in the other both genera are sister groups. Given this ambiguity, Lophotidae is considered monophyletic in this discussion.

The distinction between an almost horizontal “crest” projecting anteriorly and a more vertical and relatively shorter “crest” distinguishes †Babelichthys from †Protolophotus (see above, Taxonomic justification), but also from the extant Lophotus and the other known lophotid fossil taxa (Table 1). Conversely, in the Eumecichthys specimen that is examined, the crest is strongly projected anteriorly (angle of 72.4°) and relatively very long (Table 1). Another element is the apparent absence of vomerine fang-like teeth in †Babelichthys (it is however possible that they were present, but not preserved in the fossil), like in Eumecichthys, while they are present in Lophotus (Olney, Johnson & Baldwin, 1993). Since only one specimen is available, it is impossible to perform a thorough comparison of head morphologies at various growth stages and between individuals. Nevertheless, it seems on the basis of preserved elements that head morphology in †Babelichthys is closer to the one observed in Eumecichthys than in Lophotus, corroborating the proposition of Oelschläger (1979) that it represents a potential fossil sister group to Eumecichthys. It would then be the first known fossil unicorn crestfish. Nevertheless, †Babelichthys also differs from Eumecichthys: its crest is less strongly projecting and relatively shorter (Table 1). Moreover, no other lophotid, fossil or extant, has such an extreme enlargement and expansion of the dorsal-fin pterygiophores, the second one in particular.

Table 1 Comparison between crest measurements in selected specimens of known lophotid genera.

Species	Specimen studied	Projection of the cresta	Crest lengthb	Head lengthc	Crest length / head length	
†Babelichthys olneyi, sp. nov.	MNHN.F.EIP11	64.5°	51.5 mm	44 mm	1.17/1	
†Protolophotus elami	MNHN.F.EIP10	20°	28 mm	42 mm	0.67/1	
†Eolophotes lenis	PIN 1413/86	–17°	1.1 mm	2.1 mm	0.52/1	
†Oligolophotes fragosus	PIN 3363/121	6.7°	6.8 mm	12.7 mm	0.54/1	
Lophotus lacepede	NHMUK 1863.8.27.1	25.7°	99.8 mm	108.6 mm	0.92/1	
Eumecichthys fiski	USMN 164170	72.4°	26.8 mm	17.2 mm	1.55/1	
Notes.

a Angle (°) between the straight line from the tip of the crest to the proximal end of its anterior margin and the line drawn perpendicular to the main axis of the parasphenoid.

b Distance (mm) between the tip of the crest to the proximal end of its anterior margin.

c Distance (mm) between the anterior margin of the ethmoid and the posterior margin of the neurocranium.

The taeniosome fossil record

Taeniosome lampridiforms are known by several fossil representatives. The oldest unquestionable occurrences are all attributed to Lophotidae: the diminutive †Eolophotes lenis (Fig. 4A), from the Lutetian (Eocene) of Georgia (Daniltshenko, 1962; Daniltshenko, 1980) and †Protolophotus elami (Fig. 3), found in the same middle-late Eocene formation as †Babelichthys (see above). An additional, younger fossil lophotid is †Oligolophotes fragosus (Fig. 4B) from the early Oligocene Pshekha Formation of Adygea, northern Caucasus, Russia (Bannikov, 1999). The taeniosome fossil record also includes the trachipterid †Trachipterus mauritanicus from the Messinian (late Miocene) of Algeria (Carnevale, 2004), and a fragmentary possible oarfish (Regalecus) from the Pliocene of Italy (Bronzi, 2001; Roberts, 2012). There is no known fossil radiicephalid. Finally, the small and distinctive †Bajaichthys elegans, from the Ypresian (early Eocene) of Bolca, Italy, has been classified as a taeniosome or close relative due to its mobile jaws, elongate body and reduced caudal fin (Sorbini & Bottura, 1988; Bannikov, 2014). However, it can be confidently classified in Zeiformes, a separate teleost clade (Davesne, Carnevale & Friedman, 2017). In total, five entirely fossil taeniosome species are currently known (four Lophotidae, one Trachipteridae), a diversity expanded by the present description of †Babelichthys.

Figure 4 Other fossil taxa attributed to family Lophotidae.

(A) †Eolophotes lenis, holotype PIN 1413/86; scale bar = 5 mm. (B) †Oligolophotes fragosus, holotype PIN 3363/121; scale bars = 10 mm.

Conclusion

In the present paper, †Babelichthys olneyi, a new genus and species of Lophotidae from the Eocene of Iran is described. Few fossil representatives of Taeniosomi, an elusive group of deep-sea teleosts, are known and only one of them has been previously described in detail (Bannikov, 1999). †Babelichthys is potentially the only currently known fossil close relative of the unicorn crestfish Eumecichthys. This discovery is also significant because it expands the diversity of the middle-late Eocene Ilam fauna. Modern lophotids are found in mesopelagic environments (Olney, 2002), so the presence of at least two representatives of the family in the fauna that is mostly composed by relatives of modern deep-sea teleosts (Arambourg, 1967; Afsari et al., 2014; Přikryl, Brzobohatý & Gregorová, 2016) reinforces its potential as a valuable glimpse of the otherwise poorly known early Cenozoic deep-water ecosystems.

The author thanks Gaël Clément (MNHN) for allowing access to the specimen of study, Alexandre Bannikov (PIN) for his useful information on the lampridiform fossil record and for suggesting the redescription of this specimen, as well as Giorgio Carnevale (Università degli Studi di Torino) for insightful anatomical discussion. Sandra Raredon (USNM) kindly sent radiographs of extant lophotids for comparison, while Lilian Cazes, Philippe Loubry (MNHN) and AV Mazin (PIN) provided high-quality photographs of the fossil specimens. Finally, the editor Jérémy Anquetin and referees Alexandre Bannikov and Dave Johnson are thanked for their invaluable reviews and comments.

Institutional Abbreviations

MNHN Muséum national d’Histoire naturelle, Paris, France

NHMUK Natural History Museum, London, United Kingdom

PIN Borisyak Paleontological Institute of the Russian Academy of Sciences, Moscow, Russia

USNM National Museum of Natural History, Smithsonian Institution, Washington D.C., United States

Additional Information and Declarations

Competing Interests

Author Contributions

Data Availability

New Species Registration

The author declares there are no competing interests.

Donald Davesne conceived and designed the experiments, performed the experiments, analyzed the data, wrote the paper, prepared figures and/or tables, reviewed drafts of the paper.

The following information was supplied regarding data availability:

The measurements in Table 1 are the raw data. The material described herein, MNHN.F.EIP11 is stored at MNHN, Muséum National d’Histoire Naturelle, Paris, France.

The following information was supplied regarding the registration of a newly described species:

Publication LSID: urn:lsid:zoobank.org:pub:B677BA4F-CCF4-4678-A8A8- 502F059704D2.

Babelichthys gen. nov.: urn:lsid:zoobank.org:act:86986E5E-5FFF-465D-A987-E475FBF02966,

Babelichthys olneyi sp. nov.: urn:lsid:zoobank.org:act:D2540D1F-F169-40DE-B910-7302810615E7.

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
