# Peer review of "A fossil unicorn crestfish (Teleostei, Lampridiformes, Lophotidae) from the Eocene of Iran"

_PeerJ, doi:10.7717/peerj.3381_

## Round 0.1 · original submission · Minor Revisions

Your manuscript was reviewed by two referees who have only made a few minor suggestions for improvement. Please consult the two annotated manuscripts for these proposed changes. In addition, you will note that the first reviewer points out that the supraoccipital crest is not clearly visible in the drawing, and that the second reviewer would welcome some additional information regarding the association between dorsal fin rays and pterygiophores.

In addition to the reviewers’ suggestions, here are a few remarks I have after reading your manuscript.
- lines 23–24: you should probably cite species names, not just the genus
- line 32: ‘an anteriorly’, not ‘ a anteriorly’
- line 55: spell out genus name at the beginning of the sentence
- lines 55–57: this sentence is somewhat difficult to read, please rephrase it
- lines 65 and 70: are you sure these should be third level headings?
- line 137: ‘consists only of’ instead of ‘consists in only’
- line 210: add a comma after ‘segmentation’ to improve readability
- lines 295–311: I wonder if this part would be more appropriate in the Introduction (I let you decide)

·

Basic reporting

One reference (lines 381-382) should be corrected

Experimental design

no comments

Validity of the findings

no comments

Additional comments

Description needs in some emendation:
Lines 173-174: ascending process and alveolar process of the maxilla is a nonsense!
Line 177: unclear what is concave; not fenestra but interosseus space.
Lines 182, 209: it is not necessary what might be preserved (but not preserved).
Line 304: lampridiforms or Lampridiformes?
Line 336: add Borisyak.

·

Basic reporting

The data are well presented. I have suggested wording/English changes in the uploaded Word document. One important thing is that the serial or supernumerary association of anteriomost dorsal fin rays with relevant pterygiophores should be clarified.

Experimental design

NA

Validity of the findings

One might argue whether the separate genus is warranted, but, particlarly in the absence of a formal cladistics analysis this is a subjective matter.

Additional comments

I find very little to criticize here - just tighten up the English/wording, and it's ready to go.
I reviewed in WORD and reconverted to PDF - italicized words did not convert - sorry about that.

---

## Round 0.2 · accepted · Accept

Thank you for the revised version of your manuscript. I am now in a position to accept your submission for publication in PeerJ.

Congratulations and thank you for choosing PeerJ.